# Conserved Structure and Evolution of DPF Domain of PHF10—The Specific Subunit of PBAF Chromatin Remodeling Complex

**DOI:** 10.3390/ijms222011134

**Published:** 2021-10-15

**Authors:** Anton O. Chugunov, Nadezhda A. Potapova, Natalia S. Klimenko, Victor V. Tatarskiy, Sofia G. Georgieva, Nataliya V. Soshnikova

**Affiliations:** 1Shemyakin-Ovchinnikov Institute of Bioorganic Chemistry, Russian Academy of Sciences, 16/10 Miklukho-Maklaya St., 117997 Moscow, Russia; 2Moscow Institute of Physics and Technology (State University), 9 Institutskiy Per., 141701 Dolgoprudny, Moscow Region, Russia; 3National Research University Higher School of Economics, International Laboratory for Supercomputer Atomistic Modelling and Multi-Scale Analysis, 20 Myasnitskaya St., 101000 Moscow, Russia; 4Institute for Information Transmission Problems (Kharkevich Institute), Russian Academy of Sciences, 127051 Moscow, Russia; nadezhdalpotapova@gmail.com; 5Center for Precision Genome Editing and Genetic Technologies for Biomedicine, Institute of Gene Biology, Russian Academy of Sciences, 34/5 Vavilov St., 119334 Moscow, Russia; natasha.klmnk@gmail.com; 6Institute of Gene Biology, Russian Academy of Sciences, 34/5 Vavilov St., 119334 Moscow, Russia; tatarskii@gmail.com; 7Department of Transcription Factors, Engelhardt Institute of Molecular Biology, Russian Academy of Sciences, 32 Vavilov St., 119991 Moscow, Russia; sofia.georgieva2021@gmail.com; 8Center for Precision Genome Editing and Genetic Technologies for Biomedicine, Engelhardt Institute of Molecular Biology, Russian Academy of Sciences, 32 Vavilov St., 119991 Moscow, Russia

**Keywords:** DPF domains, PHD, PHF10, PBAF, domain evolution, duplication, H3K14ac, chromatin remodeling, genes activation

## Abstract

Transcription activation factors and multisubunit coactivator complexes get recruited at specific chromatin sites via protein domains that recognize histone modifications. Single PHDs (plant homeodomains) interact with differentially modified H3 histone tails. Double PHD finger (DPF) domains possess a unique structure different from PHD and are found in six proteins: histone acetyltransferases MOZ and MORF; chromatin remodeling complex BAF (DPF1–3); and chromatin remodeling complex PBAF (PHF10). Among them, PHF10 stands out due to the DPF sequence, structure, and functions. PHF10 is ubiquitously expressed in developing and adult organisms as four isoforms differing in structure (the presence or absence of DPF) and transcription regulation functions. Despite the importance of the DPF domain of PHF10 for transcription activation, its structure remains undetermined. We performed homology modeling of the human PHF10 DPF domain and determined common and distinct features in structure and histone modifications recognition capabilities, which can affect PBAF complex chromatin recruitment. We also traced the evolution of *DPF1–3* and *PHF10* genes from unicellular to vertebrate organisms. The data reviewed suggest that the DPF domain of PHF10 plays an important role in SWI/SNF-dependent chromatin remodeling during transcription activation.

## 1. Introduction

Gene expression is regulated by a variety of protein complexes. The chromatin epigenetic landscape, essential to the manifestation of genetic information, is shaped by histone modifications. Both individual proteins and protein complex subunits capable of interacting with chromatin possess domains that recognize different modifications of N-terminal histone tails. PHD (plant homeodomain fingers) are zinc-finger-containing domains that recognize modified N-terminal tails of H3 histones [1,2]. PHDs tandemly repeated in a face-to-back manner are referred to as DPF (double PHD finger) domains. They are organized into a single structure, which interacts with histone N-termini differently from a single PHD [3]. In mammals, DPF domains are present in six proteins, which actively participate in gene expression regulation. 

Two of these proteins, MOZ and MORF, are histone acetyltransferases, either of which can be part of the MYST complexes family acetylating H3 histone tails at lysines 9, 14, and 23 (H3K9ac, H3K14ac, and H3K23ac) [4,5]. 

Four other proteins, DPF1–3 and PHF10, are subunits of the SWI/SNF chromatin remodeling complex, which participates in replication, reparation, and transcription, changing chromatin topology and restructuring nucleosomes in the course of ATP hydrolysis. SWI/SNF complexes activate gene expression by shifting nucleosomes along the DNA strand, transferring them to another strand, removing H2A and H2B dimers, and, as a result, decreasing nucleosome density at gene promoter regions and regulatory elements, thereby activating transcription [6]. SWI/SNF complexes are important during embryonic development, as proliferation and differentiation are accompanied by changes in the patterns of intensively transcribed genes required to effectuate specific cellular functions [7,8]. 

The SWI/SNF family includes PBAF, BAF, and ncBAF complexes consisting of many proteins. These complexes possess identical cores and the ATPase subunit BRG1/BRM; however, their specific chromatin-binding modules differ. PHF10 and DPF1–3 proteins are such specific subunits of the PBAF and BAF complexes, respectively [9]. DPF domains recognize acetylated tails of histones H3 and H4, which steer the complex towards particular sites in the genome in order to remodel them. DPF domains of DPF2, DPF3, MOZ, and MORF have been crystalized, and their structure resolved [10,11,12,13,14,15,16]. Among the aforementioned six proteins, the PHF10 DPF domain possesses an amino acid sequence with the greatest level of dissimilarity to the other five. Furthermore, PHF10 functions and its role in transcriptional regulation are also unique. Mammalians have four PHF10 alternative splicing isoforms, which possess distinct domain structures and play different roles in gene transcription [17,18,19].

## 2. PHF10 Gene Is Essential for Mammalian Development and Encodes Four Evolutionary Conserved Isoforms

In mammals, the expression of PHF10 begins at the earliest stages of embryonic development: during gastrulation, PHF10 isoforms can be detected in all future organs and tissues (Figure 1) [20]. This protein is vitally important for embryogenesis, as PHF10 knockout is embryonically lethal [21]. In adult organisms, PHF10 is ubiquitously expressed [22,23], and alterations of its expression patterns correlate with the development of malignancies [24,25].

Four PHF10 transcripts are expressed starting at two different promoters and differ at the 3′-end as a result of alternative splicing. Thus, variability exists in the domain structure of their N- and C-termini (Figure 2). Each isoform is coded by its own mRNA, transcribed from the same gene—*PHF10* [17,26]. All four isoforms contain the conservatively structured SAY domain interacting with other PBAF subunits and an unstructured linker element (Figure 2) [18]. 

Two of the four PHF10 isoforms, referred to as PHF10-P (PHF10-Pl and PHF10-Ps), contain a double PHD finger domain at their C-termini. Two other isoforms, known as PHF10-S (PHF10-Sl and PHF10-Ss), include a site for phosphorylation-dependent SUMO-1 modification (known as PDSM motif) instead of the DPF domain as a result of alternative splicing. The N-terminus of the “long” PHF10-Pl and PHF10-Sl isoforms contains 46 additional amino acids which are absent in the “short” PHF10-Ps and PHF10-Ss isoforms (Figure 2). 

The domain structure of PHF10 isoforms affects their phosphorylation patterns: the long PHF10-Pl and PHF10-Sl are heavily phosphorylated at the N-terminus (N-cluster phosphorylation), while PHF10-Sl and PHF10-Ss are heavily phosphorylated at the linker domain (X-cluster). The C-terminal DPF domain prevents phosphorylation of the X-cluster in PHF10-Pl and PHF10-Ps [18]. Phosphorylation of the X-cluster substantially affects isoform stability, increasing the half-life of the PHF10-Sl and PHF10-Ss isoforms to up to 24 h, as opposed to 12 h for PHF10-Pl and PHF10-Ps [18]. PHF10 isoforms are ubiquitinylated by the b-Trcp ubiquitin-ligase, which binds to a non-canonical site within their linker domains. X-cluster phosphorylation blocks the binding of this enzyme to its recognition sites, protecting PHF10-Sl and PHF10-Ss from degradation [18].

The PBAF complex can only incorporate one of the four PHF10 isoforms, resulting in complexes with different chromatin binding capabilities [17]. Presence of a DPF domain in PHF10-Pl and PHF10-Ps enables the PBAF complex to recognize specific histone tail modifications, and, therefore, to differentially exhibit its remodeling activity. 

## 3. The Role of PHF10 Isoforms in Gene Transcription

DPF-containing PHF10 isoforms play an important role in regulating the expression of pro-proliferative and tissue-specific genes. Hyperexpression of PHF10-P in the murine cerebrum during late embryogenesis increases proliferation of the neuronal progenitor cells in basal ganglions and developing cortices, while upon knockdown of the corresponding gene the number of proliferating progenitors decreases [27]. Hyperexpression of PHF10-P, but not the PHF10-S isoforms, boosts the proliferation of HEK293T cells, which correlates with a high level of RNA polymerase II at promoters of proproliferative genes [17].

PHF10 is required for the maintenance of long-term repopulating hematopoietic stem cells [21]. PHF10-P isoforms are highly expressed in actively proliferating myelocytes [19]. Induced knockout of PHF10 in transgenic mice leads to a decrease of the myeloid progenitor pool in the bone marrow. Both the PHF10-P and PHF10-S isoforms are involved in gene expression in myeloid progenitors and their progeny (i.e., neutrophils). During the subsequent differentiation towards the myeloid lineage, expression of the PHF10-P isoforms is downregulated, while expression of the PHF10-Ss isoforms increases [19]. Differentiation is accompanied by the exit of these cells from the cell cycle and activation of the myeloid-specific genes. PHF10-P isoforms, as part of the PBAF complex, are recruited to promoter regions of neutrophil-specific surface receptor genes, which are expressed during differentiation. Following constant transcription of these genes, PHF10-P leaves gene promoters, which remain enriched in PHF10-Ss-containing PBAF complexes [19]. 

PHF10-P isoforms are thus necessary for the transcription of proliferation genes and activation of tissue-specific genes, while PHF10-S isoforms replace them to maintain the transcription of tissue-specific genes in differentiated cells.

One of the mechanisms via which PHF10 regulates pro-proliferative genes is the recently shown co-operation between PHF10 and the MYC oncogene [25]. MYC is a powerful transcriptional activator and is upregulated in 60–70% of cancers [28,29]. In A375 melanoma cells, MYC and PHF10 physically interact and regulate the expression of cell cycle genes, while lack thereof leads to cellular senescence [25]. 

The DPF-containing isoform PHF10-P thus plays an important role in activating gene transcription in response to various stimuli. PHF10-P isoforms recruitment might involve interactions with transcription activation factors such as MYC or the NF-kB dimer RelA/p50 [25,30]. It is also possible that PHF10 is recruited to gene promoters via interaction of the DPF domain with modified histone tails. 

Transcription activation is tightly connected with high levels of histone acetylation, and many transcription factors recruit complexes that modify nucleosomes. For instance, MYC interacts with the TRRAP protein, which in turn recruits histone acetyltransferases GCN5 and TIP60 [31,32]. GCN5 predominately acetylates lysines 9, 14, and 18 of histone H3 (H3K9/14/18ac), while TIP60 acetylates lysines 5, 8, and 12 of histone H4 (H4K5/8/12ac). Both MYC and RelA directly interact with and recruit the adaptor protein P300/CBP, which also acetylates histones either on its own or in the presence of its partner PCAF [33,34,35].

Lysine acetylation weakens interactions between histone tails and DNA, attracting nucleosome remodeling complexes via their subunits, which recognize such modifications [36]. The remodeling complexes then decrease nucleosome density at the promoters, allowing the RNA polymerase to effectively initiate gene transcription.

## 4. Structure of the PHF10 DPF Domain: Similar, but Different

The DPF domain is a unique protein module, wherein double PHDs are positioned in a face-to-back manner making a single structure. It interacts with histones N-tails differently as opposed to the way a single PHD does [3]. Numerous structures of DPF domains of DPF2, DPF3b, MOZ, and MORF have been determined by X-ray crystallography and NMR spectroscopy, both isolated and complexed with N-terminal peptides of histones H3 and H4 (Table 1), although no PHF10 DPF domain structure is available so far.

All DPF domains structures are very similar, revealing tandemly repeated PHDs, both containing two zinc-finger structural motifs. The first Zn^2+^ ion in each PHD is coordinated with three cysteines and one histidine, while the second one is coordinated with four cysteines. This feature is conserved in all DPF-domain-containing proteins (Figure 3A,B and Figure 4). PHD-2 is not just a copy of PHD-1: it is shorter, and its second α-helix is truncated (see comparison of secondary structure plots in Figure 3A). Complexes of DPF domains with histone H3 and H4 N-terminal peptides reveal noticeable “pockets”, suitable for three-point interaction with the peptides’ (un)modified side chains (Figure 3A). 

“Acidic pocket 1”, formed by two 100%-conserved negatively charged aspartic acid residues, serves to anchor the peptide by the positively charged arginine side chain (unmodified; H3R2 also may be methylated [39,40]). In the known structures, the salt bridge may be formed with H3R2 (MOZ [13,14,38]) and sometimes H4R17 (DPF3 [12]). 

The “hydrophobic pocket” binds the lysine side chain that is enzymatically modified by the hydrophobic moiety—H3K14ac/cr/bu or H3K9ac/me for MOZ [13,14]; and H3K14ac/cr/bu or H4K5/8/12/16ac for DPF2 and DPF3b [11,12,37]. This pocket is conserved among five DPF-containing proteins (except PHF10, see below). 

“Acidic pocket 2”, apart from anchoring another arginine residue (H3R8), preferentially binds unmethylated H3K4 and H4K20 residues, “rejecting” their trimethylated forms [41]. In known DPF domain structures, the H3K4 side chain gets into a “niche” motif [42] and forms three hydrogen bonds with the backbone oxygens of the conserved [I/M]ECK (underlined residues act as h-bonds acceptors) sequence at PHD-1 and -2 linker [14] (Figure 3A). H3K4 methylation leads to the progressive loss of these interactions, providing a basis for DPF domain preference for the unmodified H3K4 histone variant [41]. In PHF10, this pocket is also markedly different as compared to all other proteins in the family in question (see below). 

Amino acid sequence alignment taking into account the aforementioned conserved features along with their 3D representations are presented in Figure 3A,B.

The DPF domain of the PHF10 protein is not structurally characterized yet, although amino acid sequence alignment (Figure 3A) clearly reveals common ancestry and many shared features (described above). This enabled us to build a homology model of PHF10, using MOZ as the template [38] (Figure 3B) and examine its probable interaction with the H3K14cr peptide (Figure 3C). At the same time, PHF10 has the highest level of dissimilarity to other proteins within the whole family (note its early divergence from DPF1–3 and MOZ/MORF groups on the phylogenetic tree in Figure 3A), which suggests certain structural and functional differences. For the purpose of comparison of PHF10 to other proteins, a Consensus line is presented in Figure 3A, which contains not only the conserved features of the whole family, but also presumable functional distinctions of the PHF10, marked with an *arrow*. While Zn^2+^-binding sites and the acidic pocket 1 remain 100% conserved (the former is crucial for the domain structure, while the latter presumably effectuates correct histone positioning), the hydrophobic pocket and acidic pocket 2 of PHF10 exhibit distinctive differences from all the other proteins conserved within this area.

The hydrophobic pocket seems to be somehow displaced in PHF10 as opposed to other proteins. Overall, in each DPF-containing protein, the hydrophobic pocket is formed by four residues (Figure 3A,B): 100%-conserved Trp 428 and Leu 413 (PHF10 numbering) form the “bottom” of the pocket, whereas two others form the “walls”. PHF10 has two simultaneous substitutions, which shift these “walls” towards the PHD-1 half of DPF (Figure 3F). This may result in increased affinity of PHF10 to hydrophobically modified histone residues H3K14/H4K16, with a preference for bulkier ones (butyryl and crotonyl).

Acidic pocket 2 in PHF10 is more acidic than in other proteins: a distinctive pattern HHEEE emerges, which corresponds to [E/R]N[D/A]D[N/Q] in the others (Figure 3A). At the same time, this PHF10 sequence is conserved at the level of the Chordata phylum (Figure 4, blue boxes), which suggests a functional role of this pattern. Comparison of the electrostatic properties (Figure 3D,E) validates the conclusion that PHF10 may be less tolerable to lysine methylation as compared to other proteins and more selective with regard to H3K4 relative to H3K4me1/2. Experimentally, it was shown that DPF3b may bind H3K4me1, faintly binds H3K4me2 and does not bind H3K4me3 at all [43]. H3K4me1 is a well-known mark of active enhancers, and BAF gets recruited on them via recognition of these marks by the DPF3 protein [11,44]. The aforesaid peculiarity of the PHF10 acidic pocket 2 may result in active enhancers with a binding preference for BAF over PBAF complexes.

## 5. DPF Domain of PHF10 Differentiates H3K14ac and H3K4me3 Active Chromatin Marks

H3K14ac, H4K16ac, and H3K4me3 histone modifications are characteristic of actively transcribed genes [47]. By binding these chromatin modifications, DPF-containing proteins participate in nucleosome remodelling during active transcription. How do DPF domains functionally prefer binding to H3K14ac and avoid H3K4me3? 

The H3K14ac modification is found at promoters and genes coding sequences (CDS), as well as at active enhancer regions of actively transcribed genes (as shown in ChIP-seq experiments in mouse ES cells and drosophila [48,49]). H3K14ac is localized in promoters enriched in CpG islands, typical for highly expressed genes in terminally differentiated tissues, and at bivalent promoters in development genes. Some H3K14ac-enriched promoters are not transcriptionally active, but are ready to be activated by an external stimulus [48]. H3K14ac also appears to be a unique marker for genes of some metabolic pathways, G-protein coupled receptor (GPCR) signaling genes, and digestion events genes in drosophila [49]. 

H3K4me3 is a well-studied modification, localized at promoters of actively transcribed genes and sometimes at the 3′ ends; it is absent at enhancer or CDS regions [50]. Similarly to H3K14ac, H3K4me3 is associated with promoters enriched in CpG islands [51]. Despite being a marker of actively transcribed genes, H3K4me3 seems to not be required for gene activation, but rather marks stably transcribed genes [52].

We therefore hypothesize that DPF domains carrying PHF10-P isoforms in PBAF complex can selectively bind CDS, active enhancers, and some specific metabolic and digestion genes enriched in H3K14ac, and also participate in transcription activation. However, in the case of stably transcribed genes, the H3K4me3 mark prevents the DPF from interacting with nucleosomes, the implication being that PHF-P is not required to maintain transcription at this stage. Meanwhile, PBAF complexes containing PHF10-S isoforms (lacking DPF domains) interact with H3K4me3-enriched chromatin via other subunits than PHF10 and are active during constant gene transcription, but not during transcription activation. In line with this suggestion, we observed sequential replacement of PHF10-P by PHF10-S isoforms in PBAF complexes during activation of specific myeloid genes in the HL60 cells, which corroborates our hypothesis about the specialization of PBAF complexes containing PHF10-P and PHF10-S isoforms [19].

## 6. Evolution of the PHF10 Protein and Its DPF Domain

PHF10 and DPF1–3 proteins are functionally close, being subunits of chromatin remodeling complexes (PBAF and BAF, respectively). It would be worthwhile to understand whether they have a common ancestor and how they evolved. Apparently, the organism which possessed the ancestral gene of *PHF10* and *DPF1*–*3* genes group was unicellular or a simple multicellular. This conclusion can be drawn from the fact that unicellular *Naegleria* species, according to the NCBI Protein database, only has one homologous gene, while all other multicellular species carry at least two. The following scenario might be suggested: initially, a simple organism carried just one gene, the ancestor of all genes encoding DPF-containing proteins (Figure 5). Gene duplication then occurred, as a result of which two genes appeared, the *PHF10* ancestral gene and another one, ancestral to both *DPF1–3* and *MOZ/MORF* groups (Figure 5). Such a pair of genes can indeed be observed in invertebrates, for instance, in *Drosophila* and *Caenorhabditis* species. Subsequent rounds of duplication might have resulted in *DPF1-3* genes appearing, and the presence of all of them is vertebrate-specific. Most likely, two other genes, *MOZ* and *MORF*, also appeared in vertebrates (Figure 5) [53]. The appearance of PHF10-S isoforms is the most likely linked with the functional reduction of DPF occurring in transcripts and has only been detected in vertebrates. It can clearly be seen that the evolution of *PHF10* and *PHF10*-ancestor genes replicates species phylogeny (Figure 3A); however, while each DPF domain is generally conserved, their amino acid sequences differ more strongly in invertebrates. Nevertheless, residues involved in pockets formation are conserved both in vertebrates and invertebrates (Figure 4). 

Amino acid sequences of PHD-1 and PHD-2 of PHF10 are quite similar, and their tandem location most probably indicates that they are a result of domain duplication. It is difficult to say in what species domain duplication in the *PHF10* gene-ancestor took place for the first time, how gene functioning was organized in that organism, or which domain of the two, PHD-1 or PHD-2, it had initially. It might be that duplication of PHD-1 resulted in the formation of the PHD-2 domain, which is shorter in length and does not possess the long α-helix (Figure 3A), possibly due to incomplete duplication or further evolution. Or it might have been the PHD-2 domain that underwent replication at a point in time when it still carried the α-helix at that time, which was subsequently lost in the course of evolution. According to the NCBI Protein database, some species only have one domain (PHD-1 or PHD-2), whereas some have both. PHD-containing proteins are ubiquitous across all Eukaryotic supergroups [54], and specific domain structures have not been observed to belong to any certain taxonomic unit. It is quite hard to specify the time of the first duplication or pinpoint a single-PHD ancestor, but apparently the first duplication in the ancestral gene occurred in the very last eukaryotic common ancestor, not least because *Naegleria* species already have two PHDs (Figure 4). In this case all descendants had two domains, one of which might have been lost during species evolution, as occurred, as an extreme example, in plants. Another less plausible explanation is that the domain duplication in the *PHF10* gene was a commonplace event in eukaryotes in general and was triggered by different molecular mechanisms or genomics features [55,56,57]. In this scenario, the emergence of two domains in different eukaryotic species took place independently from each other and on several occasions. However, it is currently difficult to determine which of these (or possibly other) hypotheses are correct, and further studies are needed. 

## 7. Conclusions

PHF10 is a unique protein expressed as two isoforms that have the DPF domain and two isoforms without it. Isoforms are subunits of the PBAF chromatin remodeling complex and determine its nucleosome binding specificity. We performed homology modeling and determined that the PHF10 DPF domain has unique conserved structural features that enable PHF10-P isoforms to bind the H3K14ac, H3R2, and H4K16ac markers of actively transcribed chromatin, but not the functionally similar marker H3K4me3. 

The differential distribution of histone modifications across the genome determine the specificity of recruitment of PBAF complexes that contain either the PHF10-P or PHF10-S isoforms. Via the DPF domain, PHF10-P isoforms can bind CDS gene regions or enhancers, enriched in H3K14ac but depleted of H3K4me. H3K14ac appears at chromatin during gene transcription activation and attracts PHF10-P isoforms to specifically participate in the activation. H3K4me3 markers are located on genes that are already actively transcribed and are not required for the activation process. Therefore, PBAF complexes containing PHF10-S isoforms could potentially bind H3K4me3-enriched chromatin via subunits of the PBAF complex other than PHF10.

## Figures and Tables

**Figure 1 ijms-22-11134-f001:**
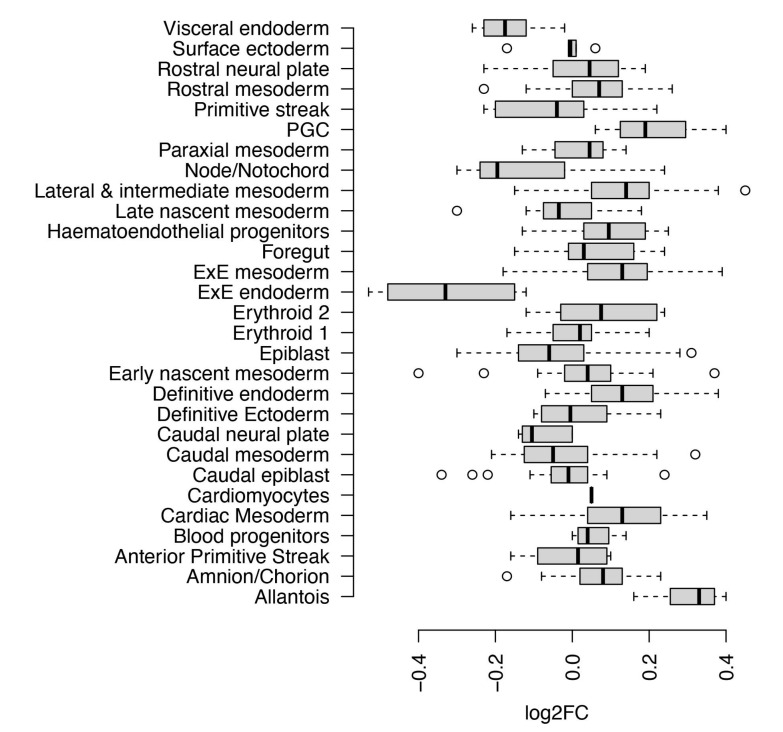
**PHF10 relative expression in different metacells in a mouse embryo during gastrulation.** Meta-cell is a transcriptional state shared by cells from numerous embryos, spanning a specific time range. Scale: log2-fold change (relative to mean expression over all metacells) Data source: [20].

**Figure 2 ijms-22-11134-f002:**
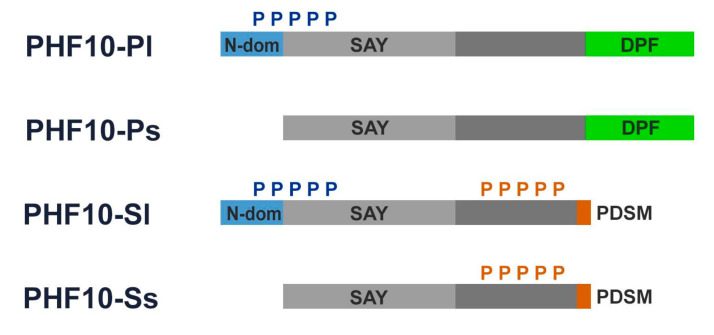
**PHF10 isoforms:***blue boxes* represent 46 amino acids at N-terminus of the long isoforms of PHF10 isoforms; *green boxes* represent DPF domains; *light grey boxes* represent SAY domains; *orange boxes* represent PDSM (phosphorylation-dependent sumoylation motif); *blue “P”* and *orange “P”* denote multiple N-terminal phosphorylation and multiple X-cluster phosphorylation, respectively.

**Figure 3 ijms-22-11134-f003:**
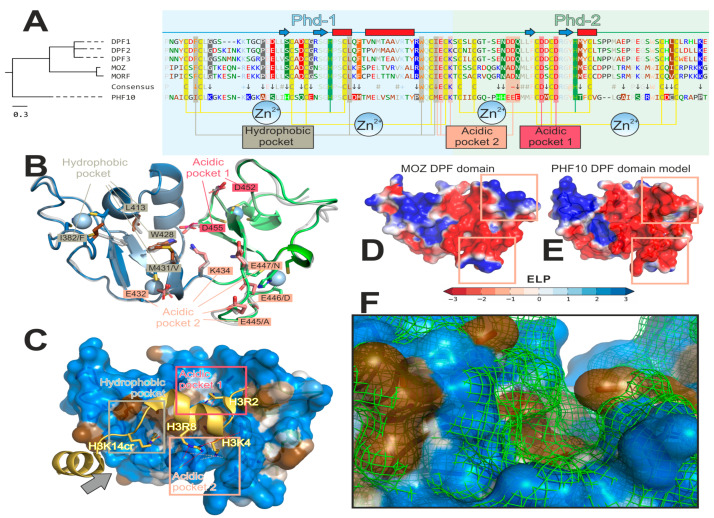
**PHF10 DPF domain has a number of peculiarities compared to other DPF-domain proteins.** (**A**) *Amino acid (a.a.) comparison suggests an early evolutionary divergence of PHF10. Left:* phylogenetic tree for six DPF-domain-containing proteins: MOZ and MORF histone acetyltransferases; DPF1–3 and PHF-10 chromatin remodeling complex subunits. *Right:* corresponding a.a. alignment formatted for better visualization of PHF10 peculiarities. PHD-1 and -2 comprising DPF-domain have *light blue* and *light green background*, respectively. Secondary structure (determined for MOZ structure (Pdb ID: 3V43) [38]) is shown above the sequences (*blue arrows* for β-strands and *red rectangles* for α-helices). A.a. are colored and grouped according to properties: *blue* (“+”)—positively charged [Lys, Arg]; *red* (“−”)—negatively charged [Asp, Glu]; *green* (“~”)—polar [Ser, Thr, Asn, Gln]; *bright-green*—His; *brown* (“#”)—aliphatic [Ala, Val, Leu, Ile, Met]; *orange*—aromatic non-polar [Phe, Tyr, Trp]; *black*—Cys, Gly and Pro. Zn^2+^-binging sites (two per one PHD module) are shown as *yellow* (Cys) and *green* (His) *vertical stripes*. A consensus line is shown between PHF10 and five other sequences. The 100%-conserved positions (with respect to the aforementioned a.a. groups) are *pale* (except Zn^2+^-binding sites and histone-binding pockets). The marked dissimilarity in the PHF10 sequence as opposed to the five other proteins is designated by an *arrow* and a *coloured background* (depending on the a.a. group) for PHF10 and other proteins (if conserved in all five resting sequences). Three histone-binding pockets are *annotated* and designated with vertical stripes: 1) hydrophobic (binds non-polar lysine modifications: H3K14ac/cr/bu in MOZ; H4S1-Nac or H4K16ac in DPF3); acidic-1 (100% conserved; anchors H3R2 in MOZ or H4R17 in DPF3); and acidic-2 (binds unmodified H3K4 or H4K20; and anchors H3R8). *B–F: Homology model of PHF10 visualizes its peculiarities in 3D* (in comparison with MOZ, which was a structural template for MODELLER 9.19). (**B**) *Overview of a model.* PHD-1 and -2 are *blue* and *green*, respectively. Parent MOZ structure is shown as a *semi-transparent gray cartoon*. Residues of the four Zn^2+^-binding sites are represented as *sticks and* ions as *spheres*. Residues of the three histone-binding pockets are *annotated*; corresponding MOZ residue is typed after a slash, if different. Note the dissimilarities in the hydrophobic and acidic-2 pockets, which may determine PHF10 specificity. (**C**) *A visualization of how PHF10 might bind the H3 tail* (based on the MOZ/H3K14cr complex (Pdb ID: 5B76) [13]). PHF10 model is shown with a *semi-transparent surface* colored as a gradient from *blue* (polar) to *brown* (non-polar) according to MHP scale [45,46]; the peptide is colored *gold*. Binding pockets are shown with *rectangles*: hydrophobic binds non-polar H3K14cr; acidic-1 anchors H3R2 via a salt bridge (*red dotted line*); acidic-2 recognizes a charged H3K4 via three hydrogen bonds with the backbone of M431, E432, and K434 (shown as *pink dotted lines*); and anchors H3R8 via salt bridge to E432 and probably E445/E447 (shown as *pink red lines*). *The gray arrow* defines a viewpoint for the F panel. (**D**,**E**): *Comparison of MOZ* (**D**) and *PHF10* (**E**) *electrostatic properties*. Note that PHF10 has a more negatively charged acidic pocket-2 due to the PHF10-specific HHEEE sequence pattern. (**F**) *Comparison of the hydrophobic pocket’s shape for PHF-10 (colored surface) and MOZ (green mesh)*. Due to simultaneous substitutions in the pocket’s walls (I382/F and M431/V; after slash is a MOZ residue), it appears to be curved in MOZ (and apparently MORF and DPF1–3), and straighter in PHF10. This may result in an increased preference of PHF10 for bulky hydrophobic modifications at H3K14 or H4K16. Interactive versions of panels B–F of this figure may be downloaded as a PyMol *.pse session from the Appendix A for this paper.

**Figure 4 ijms-22-11134-f004:**
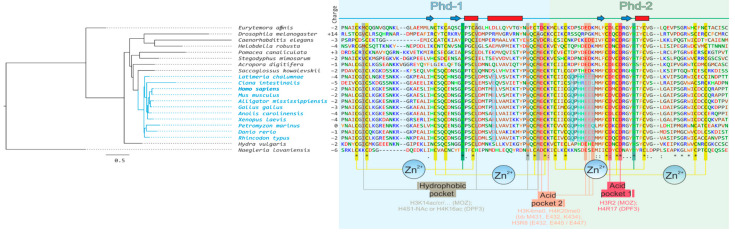
**Amino acid alignment of PHF10 from different species and (probably) several ancestral forms.** For details of the figure description, see Figure 3. Chordata in the tree are highlighted in *blue*. The conserved PHF10 features for this clade are in *blue semi-transparent boxes*.

**Figure 5 ijms-22-11134-f005:**
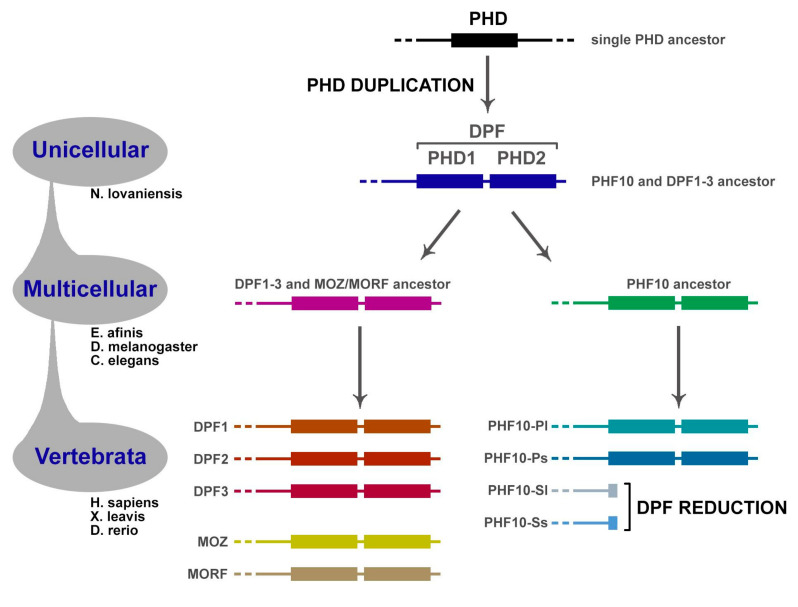
**Evolution of DPF-containing proteins.** Originally, *PHF10* and the gene ancestral to *DPF1*–*3* and *MOZ*/*MORF* groups diverged from the common ancestor. Afterwards, the latter group multiplied to five members. Most likely, PHD duplication occurred in the very last eukaryotic common ancestor, since the DPF domain is present in all these genes.

**Table 1 ijms-22-11134-t001:** DPF domains structures as of 2021.

Protein	Length	PDB	Peptide (Length)	References
DPF2	123	5VDC	-	[10]
		5B79	-	[13]
DPF3b	115	5SZB	H3K14ac (18)	[11]
		5SZC	H3K4me1K14ac (16)	
		5I3L	H3K14ac (21)	[37]
	114	2KWJ	H3K14ac	[12]
		2KWO	H4S1ac (20)	
		2KWN	H4K16ac (15)	
		2KWK	H3 wt (20)	
MOZ	131	5B75	H3K14bu (25)	[13]
		5B76	H3K14cr (26)	
		5B77	H3K14pr (25)	
		5B78	H3K14cr (25)	
	136	4LJN	-	[14]
		4LKA	H3K9ac (12)	
		4LK9	H3 wt (12)	
		4LLB	H3K14ac (15)	
	112	2LN0	-	[38]
		3V43 *	H3R2 with no modification	
MORF	111	5U2J	H3K14bu (16)	[15]
	116	6OIE	H3K14cr	[15,16]

*—This structure has maximum sequence identity (48.6% over 112 aligned positions of DPF domains) and was used as a template for homology modeling of PHF10 DPF domain in this work.

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
