# Peer review of "Conserved Structure and Evolution of DPF Domain of PHF10—The Specific Subunit of PBAF Chromatin Remodeling Complex"

_ijms, 2021, doi:10.3390/ijms222011134_

Round 1
Reviewer 1 Report
The authors provide a review of structure and function of PHF10. The review includes a lot of important details that when reorganized will add to clarity. My main suggestions are as follows:
It seems directions for creating an introduction are mistakenly added (or not removed) in the beginning of the introduction.
There are a lot of grammar, number agreement, tense and verbiage issues throughout the manuscript. In some sections, the writing does not allow the reader to follow a linear thought process.
Headings need to be reworded for clarity. Some of them do not allow the reader to understand what to expect in the information that follows. For example Heading 2 “Results PHF10 isoforms: one gene — four proteins” does not state what the authors are trying to communicate. I believe they want to communicate that the Gene PHF10 codes for four isoforms?
The flow of the paper could be improved. For example, the current flow is not linear as discussed in this sequence: the presence of 4 isoforms, isoform roles in transcription, structure, isoform function at the epigenetic level, and lastly evolution of PHF10/DPF domain. Instead, they may want to consider grouping similarly to this: 1) background, biological significance; 2) gene/protein structure; 3) function/roles in transcription & regulation, possibly relate the evolution here; 4) Conclusion to reiterate how the structure and function relationship is biologically significant.
Figure legend 1 title has a reference in it [20]. Is this indicating that data was used from this reference to generate the figure? It is unclear how this figure was generated or from what data. It would be ideal to include a description of how all figures that include data were generated.
Figure 3 legend is too long and includes information that can be simply added in a section that addresses structure-function relationships.
Author Response
The authors provide a review of structure and function of PHF10. The review includes a lot of important details that when reorganized will add to clarity. My main suggestions are as follows:
- It seems directions for creating an introduction are mistakenly added (or not removed) in the beginning of the introduction.
We removed this paragraph
2. There are a lot of grammar, number agreement, tense and verbiage issues throughout the manuscript. In some sections, the writing does not allow the reader to follow a linear thought process.
We have edited the text and hope that the new version no longer poses concerns regarding the readability of the paper.
3. Headings need to be reworded for clarity. Some of them do not allow the reader to understand what to expect in the information that follows. For example Heading 2 “Results PHF10 isoforms: one gene — four proteins” does not state what the authors are trying to communicate. I believe they want to communicate that the Gene PHF10 codes for four isoforms?
We changed some of the headings. However, we prefer not to get rid of them, as they offer a simple and obvious way to follow the principle ideas behind the review.
4. The flow of the paper could be improved. For example, the current flow is not linear as discussed in this sequence: the presence of 4 isoforms, isoform roles in transcription, structure, isoform function at the epigenetic level, and lastly evolution of PHF10/DPF domain. Instead, they may want to consider grouping similarly to this: 1) background, biological significance; 2) gene/protein structure; 3) function/roles in transcription & regulation, possibly relate the evolution here; 4) Conclusion to reiterate how the structure and function relationship is biologically significant.
We significantly reworked the text to make it more fluent and logical. We are grateful for the Reviewer's suggestions, but, upon thorough consideration , we would prefer not to change the structure, which to us looks straightforward and reasonable. Instead, to please the Reviewer, we sought the advice of a bilingual person to improve the language.
5. Figure legend 1 title has a reference in it [20]. Is this indicating that data was used from this reference to generate the figure? It is unclear how this figure was generated or from what data. It would be ideal to include a description of how all figures that include data were generated .
We added information pertaining to reference 20 in the figure legend. We plotted the PHF10 expression scheme based on data from Supplementary materials to (Mittnenzweig, M.; Mayshar, Y.; Cheng, S.; Ben-Yair, R.; Hadas, R.; Rais, Y.; Chomsky, E.; Reines, N.; Uzonyi, A.; Lumerman, L.; et al. A Single-Embryo, Single-Cell Time-Resolved Model for Mouse Gastrulation. Cell 2021, 184, 2825–2842.e22). There were no specific calculations performed to obtain the data in Figure 1.
Figure 3 contains a straightforward homology model based on the template specified and provided alignment. It is mentioned in the figure caption that Modeller software was used for modeling, and a utility called Platinum was employed to calculate the hydrophobic/hydrophilic properties of the surface of the molecules. In our opinion, there’s no need to describe this in more detail.
6. Figure 3 legend is too long and includes information that can be simply added in a section that addresses structure-function relationships.
The legend to Figure 3 is voluminous due to a large amount of information presented in this figure. Unfortunately it cannot be made any briefer, as it will no longer be exhaustive. No part of it can be moved to the main text, as the figure would not be self-sufficient anymore if that were done and the text would contain details irrelevant outside the context of the Figure.
Reviewer 2 Report
The manuscript entitled “Conserved structure and evolution of DPF domain of PHF10 – PBAF complex subunit” reviews the current information available on the double PHD finger (DPF) domain of PHF10, mainly focusing on its structure and evolution. PHF10 has peculiar DPF sequences, structure and functions, differing form other DPF-containing proteins. PHF10 is ubiquitously expressed in organisms in four isoforms differing by structure and transcription regulation functions. Despite the structure of the DPF domain has not been determined yet, the authors provided a structural model generated by homology modelling and discussed its features affecting the PBAF complex chromatin recruitment. Furthermore, the authors analyzed the evolution of PHF10 and DPF1-3 gens from unicellular to vertebrate organisms. The manuscript provides a comprehensive review on the DPF domain of PHF10, supporting its important role in chromatin remodeling during transcription activation. I would recommend the publication of this interesting manuscript after some minor revisions (listed below).
Minor revisions:
- The first 9 lines are not part of the manuscript, please remove them.
- Introduction, page 2 line7. Zinc does not require capital “Z”, change it as “zinc” here (and throughout the manuscript).
- Introduction, page 2 line 15. Change “at 9, 14 and 23 lysines” in “at lysine 9, 14 and 23”. This should be modified also in page 5 lines 5 and 6.
- Introduction, page 2 line 18. Change “by hydrolyzing ATP.” In “by ATP hydrolysis.”
- Introduction, page 2 line 24. “realization of” could be removed from the sentence.
- Title of Section 2. Remove “Results” from the title.
- Section 2, page 3, lines 7-8. The acronym “DPF” has been already introduced in the Introduction.
- Section 2 is duplicated in the manuscript, please check the section numbering.
- Section 2, page 6, line 5. “is’s” should be “it is”.
- Section 2, page 6, lines 9-10. “asparagine acid” should be “aspartic acid”.
- Section 2, page 7, line 1. Change “is not characterized structurally yet” in “is not structurally characterized yet”.
- Section 2, page 7, line 11. Change “(former” in “(the former”.
- Section 2, page 7, line 11. Change “shown DPF3b” in “shown that DPF3b”.
- Figures 3 and 4 are too small and the fonts are sometimes difficult to read (especially in Figure 4), pleas enlarge both of them (figures and fonts).
- Section 6, page 10, lines 6-7. Change “bind” in “to bind” at lines 6 and 7.
Author Response
- The first 9 lines are not part of the manuscript, please remove them.
We removed them
2. Introduction, page 2 line7. Zinc does not require capital “Z”, change it as “zinc” here (and throughout the manuscript).
We changed this
3. Introduction, page 2 line 15. Change “at 9, 14 and 23 lysines” in “at lysine 9, 14 and 23”. This should be modified also in page 5 lines 5 and 6.
We amended these
4. Introduction, page 2 line 18. Change “by hydrolyzing ATP.” In “by ATP hydrolysis.”
We amended this
5. Introduction, page 2 line 24. “realization of” could be removed from the sentence.
We removed this
6. Title of Section 2. Remove “Results” from the title.
We removed this
7. Section 2, page 3, lines 7-8. The acronym “DPF” has been already introduced in the Introduction.
We removed this
8. Section 2 is duplicated in the manuscript, please check the section numbering.
We changed the numbering
9. Section 2, page 6, line 5. “is’s” should be “it is”.
We amended this
10. Section 2, page 6, lines 9-10. “asparagine acid” should be “aspartic acid”.
We amended this
11. Section 2, page 7, line 1. Change “is not characterized structurally yet” in “is not structurally characterized yet”.
We amended this
12. Section 2, page 7, line 11. Change “(former” in “(the former”.
We amended this
13. Section 2, page 7, line 11. Change “shown DPF3b” in “shown that DPF3b”.
We amended this
14. Figures 3 and 4 are too small and the fonts are sometimes difficult to read (especially in Figure 4), please enlarge both of them (figures and fonts).
We asked the editors to make our figures larger in the final version of the article.
15. Section 6, page 10, lines 6-7. Change “bind” in “to bind” at lines 6 and 7.
We changed this
Round 2
Reviewer 1 Report
There is still some editing that needs to be done. Be sure to check all grammar and agreement issues.